# Evaluation of a Cosmetic Formulation Containing Arginine Glutamate in Patients with Burn Scars: A Pilot Study

**DOI:** 10.3390/pharmaceutics16101283

**Published:** 2024-09-30

**Authors:** HanBi Kim, InSuk Kwak, MiSun Kim, JiYoung Um, SoYeon Lee, BoYoung Chung, ChunWook Park, JongGu Won, HyeOne Kim

**Affiliations:** 1Department of Dermatology, College of Medicine, Hallym University, Kangnam Sacred Heart Hospital, Seoul 07441, Republic of Korea; khmamy1029@naver.com (H.K.); ujy0402@hanmail.net (J.U.); minggijeook@gmail.com (S.L.); victoryby@naver.com (B.C.); dermap@hanmail.net (C.P.); 2Department of Anesthesiology and Pain Medicine, Burn Center, Hallym University Hangang Sacred Heart Hospital, Seoul 07247, Republic of Korea; kwak65joy@gmail.com; 3LG Science Park R&D Center, LG Household & Healthcare (LG H&H), Seoul 07796, Republic of Korea; misunkim0407@gmail.com (M.K.); wjg8867@lghnh.com (J.W.)

**Keywords:** arginine–glutamate ion pair, post-burn scar, post-burn pruritus, skin barrier

## Abstract

**Background:** Patients with burn scars require effective treatments able to alleviate dry skin and persistent itching. Ion pairing has been employed in cosmetic formulations to enhance solubility in solvents and improve skin permeability. To evaluate the efficacy and safety of the cosmetic formula “RE:pair (arginine–glutamate ion pair)”, we analyzed scar size, itching and pain, skin barrier function, scar scale evaluation, and satisfaction in our study participants. **Methods:** A total of 10 patients were recruited, and the formula was used twice a day for up to 4 weeks. **Results:** Itching was significantly alleviated after 4 weeks of treatment (95% CI = −0.11–1.71) compared to before application (95% CI = 2.11–4.68). Transepidermal water loss (TEWL) showed an 11% improvement after 4 weeks (95% CI = 3.43–8.83) compared to before application (95% CI = 3.93–9.88), and skin coreneum hydration (SCH) showed a significant 41% improvement after 4 weeks (95% CI = 43.01–62.38) compared to before application (95% CI = 20.94–40.65). **Conclusions:** Based on the confirmation that RE:pair improves skin barrier function and relieves itching, it is likely to be used as a topical treatment for burn scars pending evaluation in follow-up studies (IRB no. HG2023-016).

## 1. Introduction

The global prevalence of burn injuries is on the rise, driven by factors such as increased industrialization, urbanization, and a higher incidence of traumatic events. In the United States, burns rank as the fourth leading cause of mortality, with approximately 2.5 million patients each year requiring medical attention. Annually, more than 100,000 burned patients are hospitalized, with 40–45% of admissions being children and 25% of these children being younger than 20 years of age. Approximately 6000 burned patients die annually, and permanent disability occurs in 50% of these patients [1,2,3]. Aside from the immediate physical trauma associated with burns, there is growing concern about the development of hypertrophic scars or keloids, which can lead to functional impairment and aesthetic challenges. Furthermore, the psychological impact of burn scars is increasingly recognized, as individuals grapple with issues such as self-esteem, body image, and social integration. Despite advances in burn care and societal progress, the incidence and severity of burns have not shown a corresponding decrease, resulting in an escalating number of patients with post-burn sequelae [4,5]. Among the various forms of discomfort, post-burn pruritus is a highly common and distressing problem affecting individuals who have suffered burn injuries [6]. Itching has been shown to significantly impact the quality of life of people with burns, causing disturbances in sleep, impairing daily activities, and affecting psychosocial well-being [7]. Post-burn itchiness may decrease over time but can persist for many years, and patients with post-burn itching often scratch their skin, resulting in more severe damage to the skin barrier [8].

Burn injuries activate a two-phase response involving both proinflammatory and anti-inflammatory processes [9]. Initially, nuclear factor κB regulates proinflammatory mediators such as TNF-α, followed by an anti-inflammatory phase characterized by increased production of anti-inflammatory cytokines and reactive species, with T helper 2 (Th2) lymphocytes and cytokines IL-4, IL-10, and TNF-α playing key roles [9]. Therefore, the development of therapies that target these intracellular inflammatory mechanisms is essential.

Following acute treatment, burned skin undergoes an acute wound-healing reaction that progresses into scar tissue over several months [10]. Itching commonly occurs during the wound-healing process following burns [11]. The aim of the following study was to improve scar quality, clinical efficacy, and patient satisfaction by applying the preparation to lesion sites of patients with second-degree (partial thickness) burns after acute treatment. The key active ingredient in the cosmetic formulation used in the present study is an ion pair consisting of arginine and glutamate. Ion pairs are utilized in cosmetic formulations to enhance solubility in solvents and improve transdermal delivery efficacy [12].

In general, well-formulated moisturizers play a crucial role in restoring barrier function to alleviate itching [13]. Using moisturizers for burn scars is crucial as they not only aid in massaging such scars but also help address the altered skin properties of scar tissue by effectively moisturizing the skin, minimizing irritation, protecting the skin barrier, and reducing transepidermal water loss (TEWL) [13]. Currently, specific recommendations for moisturizers in scar management after burn injuries are lacking and are primarily based on the clinical experience of burn specialists [14]. There have been several pilot studies evaluating the effectiveness of topical moisturizers in patients recovering from burns. For instance, the authors of a study involving acute burn patients found that using bath oil containing colloidal oatmeal resulted in significantly reduced itching and fewer requests for antihistamines from patients compared to those who utilized bath oil with liquid paraffin. Colloidal oatmeal, known for its moisturizing properties, enhances patient comfort and decreases the risk of skin damage from scratching during burn wound healing [15]. Emollients, which include basic moisturizers, Aloe vera, and coconut oil, are important for softening the outermost layer of dry skin (stratum corneum) and are critical in healing burn wounds by improving skin barrier function [16]. They are recognized for their role in minimizing sensitivity and promoting healing [17]. The authors of several studies have investigated various moisturizers such as water-based creams, wax and herbal oil creams, silicone creams, paraffin/oil/mineral oil products, and Aloe vera gels in managing scar formation, skin hydration, and TEWL [18]. However, identifying the optimal moisturizer for managing post-burn scarring remains elusive due to the complexity of scar healing and the multifaceted nature of scar assessment [19].

Currently, various moisturizers are employed to treat burn scars, including Pantenol-ratiopharm^®^ (Ratiopharm GmbH, Ulm, Germany), Alhydran^®^ (Asclepios GmbH, Breisgau, Germany), and Theresienöl^®^ (Theresienöl GmbH, Kufstein, Austria). Pantenol-ratiopharm^®^ is an occlusive ointment used for post-burn scar care, focusing on minimizing TEWL. Its key component, Dexpanthenol, is renowned for its wound-healing properties [20,21]. Alhydran^®^ is an Aloe vera-based moisturizer frequently employed in natural medicine, primarily for skin hydration and scar management. Notably, it has shown efficacy, particularly when used in gel form, in treating wounds [22,23,24]. Therrienöl^®^ is an ointment consisting of apple and lily extracts, along with tocopheryl acetate (vitamin E), which is recognized for its ability to promote cell repair and the regeneration of damaged skin [25]. Nevertheless, the ingredients and characteristics of topical agents used to treat post-burn scars have yet to be standardized. While there is high demand for specialized cosmetic formulations to treat burn scars, research on topical agents designed to alleviate symptoms and signs associated with these scars remains limited.

The authors of several studies have highlighted the effectiveness of arginine and glutamine in wound healing and improvement. L-arginine, a basic amino acid metabolized by arginase I or nitric oxide synthase II (NOS II), is recognized for its role in cell division, collagen synthesis, wound healing, and immune homeostasis [26,27]. L-arginine is known for its ability to reduce Th1 cytokine release and enhance Th2 cytokine production during severe burn infections, suggesting its immunomodulatory properties [28]. It also exhibits therapeutic effects on burns and UV-induced erythema [29,30]. Biologically, L-arginine supplementation may induce adenosine monophosphate (AMP) dephosphorylation, increasing intracellular adenosine triphosphate (ATP) regeneration, which plays a central role in energy homeostasis [30]. Furthermore, L-arginine is known to enhance blood circulation, which can help alleviate itching, as itching may be caused by issues related to blood circulation [16].

Glutamate has been shown to promote keratinocyte proliferation and plays a crucial role in skin metabolism and function. It is involved in collagen biosynthesis in fibroblasts and also plays significant roles in the central nervous system, contributing to the growth of both keratinocytes and fibroblasts. [31,32,33]. In the epidermis, glutamate contributes to the formation of the cornified envelope (CE), which acts as a barrier through the action of transglutaminase enzymes [34]. In addition, glutamate is reported to regulate the secretion of GRP, a key neurotransmitter involved in itch transmission, and is known to have therapeutic effects in treating post-burn injuries [35,36].

Ion pairs, formed by combining salts with opposite surface charges, are commonly used to enhance solubility and skin permeability, thereby improving transdermal delivery [12]. Glutamate’s low solubility in water (7.5 g/L at 20 °C) limits its use in high-concentration cosmetics or ointments [37]. To overcome this limitation, glutamate’s solubility can be increased through the formation of ion pairs with other amino acids [38]. Additionally, L-arginine not only significantly increases the percutaneous absorption rate through the formation of strong intermolecular bonds with counter ions, but also greatly enhances skin penetration due to its high affinity for free fatty acids in the skin [39,40].

Based on the characteristics of L-arginine and glutamic acid, a cosmetic formulation combining ion pairs (known as RE:pair) was developed [41]. The results of our previous study demonstrated that RE:pair provides superior therapeutic effects compared to individual amino acids or simple mixtures by reducing wound area, restoring barrier function, and improving skin elasticity, confirming its potential as an effective cosmetic ingredient for wound healing and enhancing skin elasticity [41]. The purpose of the experiment presented herein was to evaluate the effect of cosmetic preparations containing arginine–glutamate ion pairs on burn scar severity, post-burn dryness, itching, pain, and patient satisfaction among individuals who have undergone acute treatment for partial-thickness burns.

## 2. Materials and Methods

### 2.1. Study Design

In our previous report, detailed methods for producing the formulations are described [42]. Briefly, deionized water was added to arginine (Sigma-Aldrich, St. Louis, MO, USA) and glutamic acid (Sigma-Aldrich, Inc.) and stirred at 800 rpm and 50 °C for approximately 20 min until a homogeneous and uniform liquid was formed without precipitation, achieving a molar ratio of 1:1. After precipitating RE:pair through the addition of methanol, the residual methanol was removed via distillation under reduced pressure and filtration. This process was repeated three times to prepare RE:pair, an oil-in-water (O/W)-based cosmetic formulation containing 1.8% of an ionic pair of arginine and glutamate at a molar ratio of 1:1. The main oil used was caprylic/capric triglyceride.

To conduct a pilot study on the evaluation of RE:pair’s ability to alleviate post-burn scars, we recruited 10 patients (8 male (80%); 2 female (20%); average age 52.7 ± 12.28 years) aged 18 or older with post-burn scars who were undergoing follow-up (F/U) after primary treatment at the Burn Center of Hallym University Hangang Sacred Heart Hospital. Before applying the cosmetic formulation, the patient’s post-burn scars were scored as “mild to severe” based on the Vancouver Scar Scale (VSS) score. Treatment and F/U evaluations were conducted between May and December 2023.

All subjects provided informed consent for inclusion before participation in the study. Age, sex, and duration of scarring were collected as variables for each patient. The study was conducted according to the principles expressed in the Declaration of Helsinki, and the clinical trial was approved by the Institutional Review Board of Hallym University Hangang Sacred Heart Hospital (IRB no. HG2023-016). Written informed consent was obtained from all study participants.

Inclusion and exclusion criteria were established as follows:

Inclusion criteria:(1).Individuals who have completed acute burn treatment at a dermatology or plastic surgery clinic.(2).Individuals who have voluntarily provided informed consent after receiving sufficient explanation about the purpose and details of the study.(3).Individuals who understand the nature of the clinical trial, are cooperative, and can participate until the trial’s completion.(4).Individuals who can be reliably followed up during the examination period.

Exclusion criteria:(1).Individuals with acute inflammatory skin disease or moderate to severe symptomatic infection at the site of observation.(2).Individuals with uncontrolled systemic or chronic disease.(3).Individuals who have undergone immunotherapy or biologics (including oral corticosteroids) within 4 weeks prior to enrollment.(4).Individuals who have used topical corticosteroids or topical immunosuppressants within 1 week prior to enrollment.(5).Individuals who are required to take prohibited drugs as part of this clinical trial.

Registered clinical trial subjects were provided with the test preparation on the date of the clinical trial (Visit 1; Baseline). The clinical trial subjects applied the RE:pair formula twice a day to the burn area. If necessary, sunscreen could be used, but hot baths were prohibited; showers were permitted (within 15 min) and the product was reapplied immediately afterward. The clinical trial subjects visited the institution for evaluation at Week 2 (Visit 2; Day 14 ± 3) and Week 4 (Visit 3; Day 28 ± 3).

### 2.2. Outcome Measures

Clinical evaluations involving the use of photographs were conducted at baseline, Week 2, and Week 4. To assess efficacy, itching, pain, TEWL, skin corneum hydration (SCH) levels, and VSS scores were measured at the lesion site using the Tewometer^®^ TM300 probe (Courage & Khazaka GmbH, Cologne, Germany) and the Corneometer^®^ CM825 probe (Courage & Khazaka GmbH, Cologne, Germany). Two blinded and trained dermatologists evaluated the scores using clinical photos. The devices were placed perpendicularly on the skin surface within 2 min of each measurement, and TEWL and SCH were measured.

Patients were evaluated using a subjective scale ranging from 0 to 10 to assess itching and pain. Additional clinical assessments were based on the VSS, which comprises four components: pigmentation, thickness (rated from 0 to 3), erythema, and pliability (rated from 0 to 5). A score of zero indicates normal skin, while higher scores indicate increasing severity at the lesion site. These assessments were conducted under constant environmental conditions of temperature (20–24 °C) and humidity (28–38%) for a duration of 2 min.

### 2.3. Statistical Analysis

Statistical analyses were performed using IBM SPSS Statistics 27.0 software (IBM, Armonk, NY, USA) and R software version 3.5.1 (R Foundation for Statistical Computing, Vienna, Austria). Categorical variables are presented as frequencies and percentages, while continuous variables are expressed as the mean ± standard deviation (SD). Unless otherwise stated, all statistical tests were two-sided tests with a significance level of α = 0.05.

## 3. Results

### 3.1. Baseline Characteristics of the Patients

Ten patients were enrolled in this study (Table 1). The initial severity of burn scars among patients before application ranged from mild to severe based on the VSS score. The mean length of the lesional site was 7.81 cm, with 13 out of 17 lesions classified as severe (≥5 cm). No significant associations were found in the repeated-measures ANOVA (RM-ANOVA) test, which assessed sex, age, and duration of burn scar as variables.

### 3.2. Efficacy of RE:pair in Relieving Itching and Pain

The severity of post-burn scars among the study participants ranged from mild to severe. Itching was assessed using a scale from 0 to 10 among those patients who reported experiencing this issue, allowing us to observe trends in reduced itching over time. The chemical structure of arginine glutamate was retrieved from ChemSpider (ChemSpider ID: 144883) and is illustrated in Figure 1.

A principal component analysis (PCA) plot demonstrated clinical score differences between before (Week 0) and after (Week 4) application (Figure 2a). The numerical rating scale (NRS) itching score decreased from an average of 3.4 ± 2 before application (95% confidence interval (CI) = 2.11–4.68) to 1.8 ± 2 at Week 2 (95% CI = 0.49–3.10) and further to 0.8 ± 1.47 at Week 4 (95% CI = −0.11–1.71), indicating reduction of 47% (*p* = 0.07) after 2 weeks and 76% (*p* = 0.004) at Week 4. Specifically, the patient with the highest itching score of 7 before application showed a notable reduction to 3 points after 4 weeks of treatment (Figure 2b). Post hoc testing using Bonferroni’s method confirmed a significant decrease in itching after 4 weeks (*p* = 0.014).

Regarding pain, almost none of the patients experienced this issue before application, with only one patient reporting a score of 3 points. This score decreased to 0 points after application.

### 3.3. Changes in Skin Barrier Function and VSS with RE:pair

TEWL was measured to assess changes in skin barrier function. We aimed to obtain more reliable experimental results by comparing measurements from the normal skin area and the burn scar area. In the normal area, TEWL values were recorded as 4.51 ± 2.56 before application, 5.02 ± 4.38 at Week 2, and 4.62 ± 2.87 at Week 4, showing no significant change compared to before application.

The average TEWL values in the scar area were 6.91 ± 4.79 before application (95% CI = 3.93–9.88), 6.21 ± 4.13 at Week 2 (95% CI = 3.64–8.77), and 6.13 ± 4.35 at Week 4 (95% CI = 3.43–8.83). These results represent an 11% improvement after 4 weeks of application; however, no significant difference was observed compared to before application (Figure 3a).

When measuring the effect on each area, TEWL values for the arms and hands were 8.03 ± 5.90 before application (95% CI = 2.25–13.8), 5.98 ± 3.26 at Week 2 (95% CI = 2.78–9.18), and 4.99 ± 4.92 at Week 4 (95% CI = 0.16–9.81). These results indicate 25% improvement after 2 weeks and 37% improvement after 4 weeks of application, although these changes were not statistically significant. For the thighs and feet, an 11% improvement was observed after 4 weeks compared to before application; however, statistical significance was not reached (Appendix A).

We also evaluated efficacy by comparing SCH values between the normal area and the burn scar area. In the normal area, SCH values were 36 ± 17.47 before application, 34.9 ± 13.31 at Week 2, and 43.3 ± 8.88 at Week 4, showing no significant difference compared to before application.

In the scar area, SCH values were 30.8 ± 15.89 before application (95% CI = 20.94–40.65), 48.1 ± 19.18 at Week 2 (95% CI = 36.21–59.98), and 52.7 ± 15.62 at Week 4 (95% CI = 43.01–62.38). These findings represent an increase of 56% (*p* = 0.03) after 2 weeks and 71% (*p* = 0.03) after 4 weeks compared to before application. Post hoc testing using Bonferroni’s method confirmed a significant increase in SCH values after 4 weeks of application (*p* = 0.032) (Figure 3b).

When measuring the effect on each area, the arms and hands showed SCH values of 36.25 ± 19.99 before application (95% CI = 16.67–55.84), 47.75 ± 11.88 at Week 2 (95% CI = 36.10–59.39), and 53 ± 11.46 at Week 4 (95% CI = 41.76–64.23), indicating a 46% increase after 4 weeks compared to before application; however, this change was not statistically significant. In contrast, for the thighs and feet, SCH values were 27.5 ± 12.41 before application (95% CI = 15.32–39.67) and 61.5 ± 9.06 after 4 weeks (95% CI = 52.61–70.38), demonstrating a significant improvement of 123% (*p* = 0.02) after 4 weeks of application compared to before application (Appendix A).

### 3.4. Changes in Burn Scar Characteristics with RE:pair

To evaluate the subjects’ VSS scores, their pigmentation, thickness (height score), erythema, and pliability were assessed numerically. Pigmentation decreased on average by 8.3% after 4 weeks (95% CI = 0.90–1.29) compared to before application (95% CI = 0.93–1.46). Regarding thickness (height score), almost all patients did not have a score before application, and among those who did (n = 2), their scores did not change after application. Erythema showed a tendency to decrease by 21% after 4 weeks (95% CI = 0.90–1.29) compared to before application (95% CI = 1.07–1.72). Pliability also decreased by an average of 13.63% after 4 weeks (95% CI = 1.70–2.09) compared to before application (95% CI = 1.80–2.59). When calculating the four items using the VSS, there was an average decrease of 5.88% after 2 weeks (95% CI = 3.98–5.61) compared to before application (95% CI = 4.20–5.99) and an average decrease of 15.68% after 4 weeks (95% CI = 3.64–4.95). However, no statistical significance was observed (Figure 4a).

Figure 4b shows the condition of a burn scar area before and after the application of the moisturizer. There is a noticeable reduction in redness and an overall improvement in skin condition.

## 4. Discussion

As described above, we conducted a pilot study to evaluate the clinical effectiveness of RE:pair, a formulation containing arginine–glutamate ion pairs in alleviating post-burn scars.

The improvement in NRS itching scores in the scar area following application was evaluated. A significant decrease in itching score was observed after 4 weeks (95% CI = −0.11–1.71) compared to before application (95% CI = 2.11–4.68) while statistical significance was not observed for pain, as nearly all patients did not report pain before application; there was a trend toward reduction. Therefore, this formula can be considered effective in alleviating pruritus in post-burn patients.

TEWL and SCH measurements were used to evaluate skin barrier function, with results shown in Figure 3. TEWL was used to assess the integrity of the skin barrier to identify potential damage or dysfunction. Elevated TEWL values suggest impaired skin barrier function, leading to increased water loss [42]. By comparing the normal and burn scar areas, more reliable experimental results were obtained. While no significant differences were found in normal areas, a noticeable improvement was observed in the scar areas. Further analysis, separating the results by arms and legs, confirmed that skin barrier function improved after both 2 and 4 weeks of application.

To assess clinical improvement, the VSS was employed to evaluate pigmentation, thickness, erythema, and pliability. Although there was a general decrease in VSS scores compared to before application, the changes did not reach statistical significance. This may be due to the small sample size and inherent limitations in evaluating burn scar characteristics. Detecting significant changes with topical agents alone can be challenging, as burn scarring typically stabilizes over time. The VSS, which assesses scar color and thickness, aligns with our study’s results and is consistent with previous research on burn scarring [43,44]. Nonetheless, significant improvement in skin condition was observed in the burn scars following the application of the moisturizer.

In the evaluation of the subjects’ overall satisfaction with the application area, all evaluators rated it as “excellent” both 2 weeks and 4 weeks after application compared to before application. Continuous use showed highly satisfactory improvement in patient symptoms. The RM-ANOVA test found no significant associations among ‘sex, age, and duration of burn scar’, which were expected variables, and the evaluation items.

In conclusion, this study measured improvements in itching, pain, TEWL, and skin moisture, demonstrating the potential efficacy of cosmetic formulations containing arginine–glutamate ion pairs for managing post-burn scars. Our results confirm the formulation’s positive effects on patient outcomes. The high satisfaction scores among study patients underscore the acceptability and usability of the product in clinical settings.

As this study is a pilot investigation aimed at identifying initial clinical effects and considering that the moisturizers used contained a variety of cosmetic ingredients in addition to the active components, it remains unclear whether the active ingredients specifically contributed to improving skin barrier function and alleviating burn-induced itching. Aside from the active ingredients, the formulation included components such as 1,3-butylene glycol and 1,2-hexanediol, which act as moisturizers, solvents, and preservatives; cyclopentasiloxane and dimethicone, which are silicones that enhance the stability of the emulsion; polysorbate, which is an emulsifier; and tromethamine, which is used as a pH adjuster [41].

Moving forward, we intend to include diverse patient populations, extend the study duration to longer periods, and explore the extended benefits of arginine–glutamate ion pairs in our future research endeavors. Additionally, plans are underway for a study to investigate potential synergies and underlying mechanisms by combining this approach with other scar management modalities. This approach aims to provide a more comprehensive understanding of the therapeutic effects of the product.

## 5. Conclusions

As outlined above, we conducted a pilot study to evaluate the clinical efficacy of RE:pair, an arginine–glutamate ion pair formulation for improving post-burn scars Our findings indicate that the ion pair was more effective than the individual amino acids, as corroborated by previous studies [12,41]. The treatment led to reductions in itching and pain, improved skin barrier function (evidenced by decreased TEWL and increased SCH), and high levels of patient satisfaction. Although participants’ VSS scores generally decreased, statistical significance was not achieved due to the small sample size inherent to pilot studies. Future research will explore the fundamental mechanisms in more depth, include a larger sample size, and incorporate a placebo group to examine the long-term effects of the treatment. Additionally, we plan to investigate potential synergistic effects in scar treatments for various patient populations, beyond just burn patients.

## Figures and Tables

**Figure 1 pharmaceutics-16-01283-f001:**
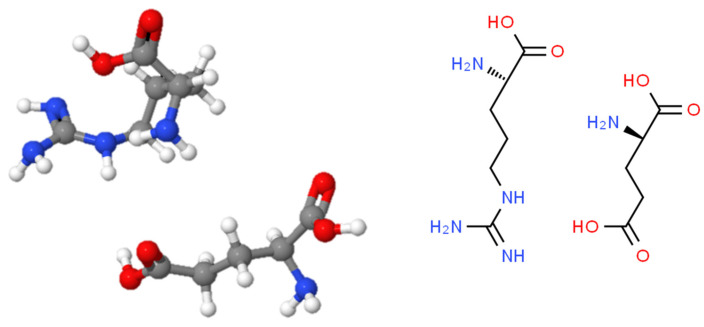
The chemical structure of arginine glutamate (ChemSpider ID: 144883) obtained from ChemSpider.

**Figure 2 pharmaceutics-16-01283-f002:**
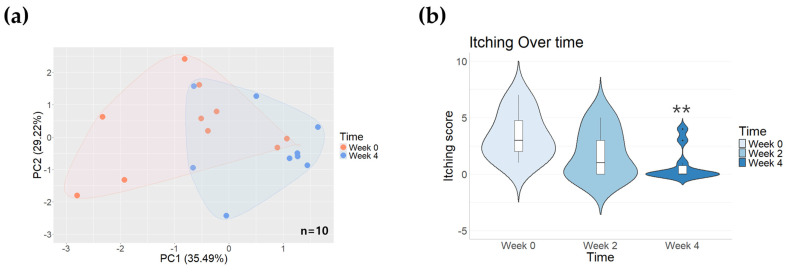
Changes in itching score at the lesional site before and after treatment (n = 10). Data are represented as the means ± standard deviation (SD). (**a**) The PCA plot shows several clinical score differences between before and after application. Numerical rating scale (NRS) for (**b**) itching score (** *p* < 0.01 vs. Week 0).

**Figure 3 pharmaceutics-16-01283-f003:**
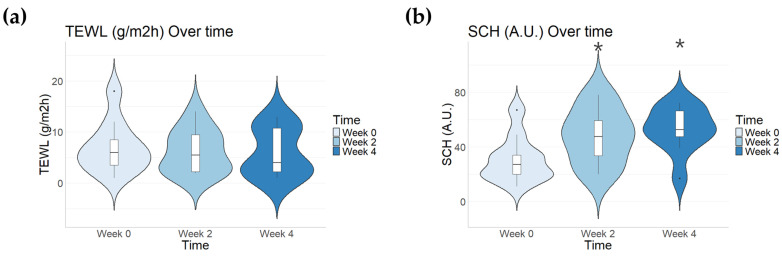
Changes in TEWL and SCH values after 2 weeks and 4 weeks of treatment (n = 10). Data are represented as the mean value ± SD. (**a**) TEWL (g/m^2^h) and (**b**) SCH (A.U.) scores (* *p* < 0.05 vs. Week 0).

**Figure 4 pharmaceutics-16-01283-f004:**
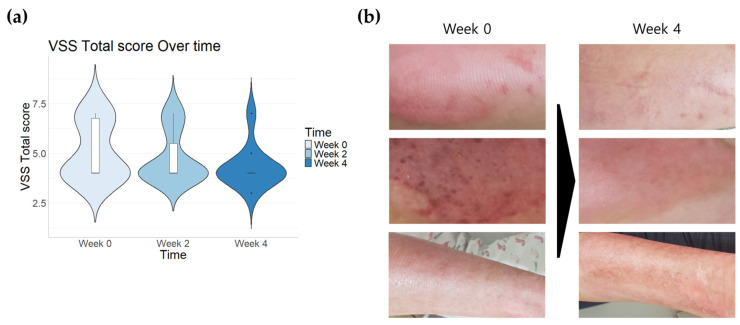
Changes in total Vancouver Scar Scale (VSS) score and surface appearance after 2 weeks and 4 weeks of treatment (n = 10). Data are represented as the mean value ± SD. (**a**) VSS total score and (**b**) photographs of the scar area.

**Table 1 pharmaceutics-16-01283-t001:** Demographic data of the subjects (n = 10).

Variables		Total (n = 10)
Age, Mean ± SD		52.7 ± 12.28
Sex, n (%)	Male	8 (80%)
	Female	2 (20%)
Application site	Arm + hand	4 (50%)
	Thigh + foot	4 (30%)
	Trunk	2 (10%)
Baseline transepidermal water loss (TEWL), mean ± standard deviation (SD) (normal)		4.51 ± 2.56
Baseline TEWL, mean ± SD (lesional)		6.91 ± 4.79
Baseline skin hydration, mean ± SD (normal)		36 ± 17.47
Baseline skin hydration, mean ± SD (lesional)		30.8 ± 15.89

## Data Availability

Individual subject data are not publicly available due to the requirement to uphold ethical standards and safeguard the confidentiality of the subjects. The data supporting this study’s findings are available from the corresponding author upon reasonable request.

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
