# Peer review of "Evaluation of a Cosmetic Formulation Containing Arginine Glutamate in Patients with Burn Scars: A Pilot Study"

_pharmaceutics, 2024, doi:10.3390/pharmaceutics16101283_

Round 1
Reviewer 1 Report
Comments and Suggestions for Authors
In the current study by Kim et al., the authors worked on a pilot study to assess the clinical effectiveness of RE:pair, an arginine glutamate ion pair formulation, in alleviating post-burn scars. This is a follow up study on their previous observations showing RE:pair improved wound healing skin elasticity. In the present study, the authors tested RE:pair on lesion site of patients with second-degree burns after the acute treatment and showed that use of RE:pair improves skin barrier function and significantly relieves itching and propose RE:pair can be likely to be used as a topical treatment for burn scars. However, additional clinical assessments including pigmentation, thickness, erythema, and pliability based on the Vancouver scar scale showed no significance.
This manuscript can be benefited from addressing the following points.
1. It was shown in the previous study that RE:pair exerts its therapeutic action on wound healing and skin elasticity improvement by promoting the proliferation of keratinocytes and enhancing collagen synthesis in fibroblasts. What is the mechanism of RE:pair in relieving itch? Itch is a paradigm of neuroimmune crosstalk. PMID 32673566 have shown that TGF-β induced IL-31 expression from dermal DC activates sensory neurons and stimulate wound itching. Similarly, PMID 34986325 and 36604775 have shown that Glutamate acts as a key neurotransmitter for itch. It would help if authors discussed on these lines detailing the possible mechanisms of action.
2. It would be ideal if the authors can provide the images used for Clinical evaluations.
3. Given the minimal sample size, additional readings may strengthen authors observation.
4. Supplementation with arginine, glutamine, omega-3 fatty acids, vitamins, and trace minerals, were shown to improve wound healing process and treatment efficacy. What is the role of arginine and glutamine on their own in relieving itch? Would combination of arginine or glutamine with other established components in wound healing will enhance relieving itch? It would be ideal if the authors make a comparative study and provide a rational for RE:pair as a better choice!
Comments on the Quality of English LanguageUse of language can be improved
Author Response
Thank you very much for your review.
We have compiled the responses to your comments in a Word file, which we have attached.
We also received English editing as you recommended.

Reviewer 2 Report
Comments and Suggestions for Authors
The current study assessed Formulation Containing Arginine Glutamate in Patients with Burn Scars
The title and abstract are relevant to the text's subject. The writing is clear and the article is well-written. The methods are clearly described, and the results and discussion are supported by enough references and statistical analysis.
However, several points needed to be addressed by authors to improve the quality of the paper.
Comments:
Lines 140-141: If necessary, sunscreen can be used, but hot baths were prohibited; showers were permitted (within 15 minutes), and the product was applied immediately
Did the participants apply sunscreen? If so, how did the authors assess its impact?
Why was having a hot bath forbidden?
Did any particular shampoo get used by the participants during the shower? Were all of them using the same shampoo?
Were any of the individuals using moisturizers?
Lines 271 to 304: Both of these paragraphs are more of an overview of the literature than a discussion.
Lines 311 to 334: The results that have already been mentioned by the authors are frequently repeated; these should not be repeated, and the results should be explained in this section
Comments on the Quality of English LanguageNA
Author Response

(The authors gave the same response as above.)

Reviewer 3 Report
Comments and Suggestions for Authors
Please see the attached file.

Extensive language editing is essential.
Author Response

(The authors gave the same response as above.)

Reviewer 4 Report
Comments and Suggestions for Authors
Patients with burn scars require effective treatments that can alleviate dry 16 skin and persistent itching. Ion pairing has been employed to enhance solubility and improve skin permeability.
To evaluate the efficacy and safety of the cosmetic formula “RE:pair (arginine glu- 18 tamine ion pair)”, the authors analyzed scar size, itching and pain, skin barrier function, scar scale evaluation, and satisfaction. The results show that RE:pair improves skin barrier function 26 relieves itching, it is likely to be used as a topical treatment for burn scars, pending follow-up studies. The manuscript is interesting but is necessary to implement wind two kinds of informations: a) References more appropriate about cellular mechanisms of burns b) Pcutres of burns before and after the treatmen with legends Comments on the Quality of English LanguageModerate editing of English language required
Author Response

(The authors gave the same response as above.)

Round 2
Reviewer 3 Report
Comments and Suggestions for Authors
Please see comments in the manuscript.
This is indeed an improvement on the paper, well done.

There are still some small technical issues that could be addressed.
Author Response
Thank you very much for your feedback.
We have addressed your comments in the attached Word document.

Reviewer 4 Report
Comments and Suggestions for Authors
The authors have asked correctly to my questions
Comments on the Quality of English LanguageModerate editing of English language required.
Author Response
Thank you very much for your feedback.
We have completed the English editing as you suggested.